# Tumor Microenvironment Regulation and Cancer Targeting Therapy Based on Nanoparticles

**DOI:** 10.3390/jfb14030136

**Published:** 2023-02-28

**Authors:** Shulan Han, Yongjie Chi, Zhu Yang, Juan Ma, Lianyan Wang

**Affiliations:** 1School of Pharmaceutical Sciences, Jilin University, Changchun 130021, China; 2Key Laboratory of Green Process and Engineering, State Key Laboratory of Biochemical Engineering, Institute of Process Engineering, Chinese Academy of Sciences, Beijing 100190, China; 3University of Chinese Academy of Sciences, Beijing 100049, China; 4Department of Clinical Laboratory Medicine, Beijing Shijitan Hospital, Capital Medical University, Beijing 100038, China

**Keywords:** antitumor therapy, tumor, microenvironment regulation, nanoparticles

## Abstract

Although we have made remarkable achievements in cancer awareness and medical technology, there are still tremendous increases in cancer incidence and mortality. However, most anti-tumor strategies, including immunotherapy, show low efficiency in clinical application. More and more evidence suggest that this low efficacy may be closely related to the immunosuppression of the tumor microenvironment (TME). The TME plays a significant role in tumorigenesis, development, and metastasis. Therefore, it is necessary to regulate the TME during antitumor therapy. Several strategies are developing to regulate the TME as inhibiting tumor angiogenesis, reversing tumor associated macrophage (TAM) phenotype, removing T cell immunosuppression, and so on. Among them, nanotechnology shows great potential for delivering regulators into TME, which further enhance the antitumor therapy efficacy. Properly designed nanomaterials can carry regulators and/or therapeutic agents to eligible locations or cells to trigger specific immune response and further kill tumor cells. Specifically, the designed nanoparticles could not only directly reverse the primary TME immunosuppression, but also induce effective systemic immune response, which would prevent niche formation before metastasis and inhibit tumor recurrence. In this review, we summarized the development of nanoparticles (NPs) for anti-cancer therapy, TME regulation, and tumor metastasis inhibition. We also discussed the prospect and potential of nanocarriers for cancer therapy.

## 1. Introduction

The data from the International Agency for Research on Cancer show that cancer is a major disease with the highest morbidity and mortality worldwide [1]. Common clinical anti-cancer treatments, such as radiotherapy, chemotherapy, surgery, and other targeted cancer treatments, have many shortcomings including multiple complications, high rate of recurrence and metastasis, off-target effect, easy drug resistance, and serious toxicity, which greatly reduce the patients’ quality of life [2,3,4,5]. Cancer immunotherapy is emerging as the fifth anti-cancer treatment strategy. The treatment activates the host immune system for recognizing and destroying cancer cells in an antigen-specific manner [6]. In recent years, some immunotherapy, such as the programmed cell death protein 1(PD-1), cytotoxic T lymphocyte-associated protein 4 (CTLA-4), and the chimeric antigen receptors T cells (CAR-T), have achieved promising clinical therapeutic effects in cancer. However, overall treatment response rates still remain low (<20%), which is far below expectations if cancer types are taken into account as a whole [7,8,9]. 

Increasing evidence indicates that the complexity of TME results in poor treatment effectiveness of cancer immunotherapy, chemotherapy, and targeted therapy [10,11,12]. The immunosuppressive microenvironment supports the occurrence and development of the tumor, which leads to immune escape of tumor cells [13]. The tumor microenvironment (TME) includes diverse immunosuppressive factors, such as incapacitated immunostimulatory cells (e.g., dendritic cells (DCs), T helper (Th) cells, cytotoxic T lymphocytes (CTLs), M1-like tumor-associated macrophages (M1-like TAMs), natural killer (NK) cells), activated immunosuppressive cells (e.g., myeloid-derived suppressor cells (MDSCs), M2-like tumor-associated macrophages (M2-like TAMs), regulatory T cells (Treg), cancer-associated fibroblasts (CAFs), the tumor vasculature, extracellular matrix (ECM), hypoxia, and low pH [14,15,16,17,18]. The above immunosuppressive factors contribute to a variety of mechanisms for tumor therapy inhibition in tumor-associated macrophages (TAMs), which further promote the occurrence and development of cancer [19,20]. Therefore, it is necessary for improving anti-cancer treatment through regulation of TME immunosuppression.

Nanomaterials are widely employed for anti-tumor treatment and TME regulation [21,22,23]. The nanomaterials with modulators such as multifunctional platforms can effectively eliminate the primary cancer, inhibit the distal metastasis, and prevent cancer recurrence [22,24,25,26]. Nanomaterials mainly regulate TME through the following four mechanisms: (i). Promote the immunogenicity of cancer antigens [27,28,29,30,31]; (ii). Activate disabled immune cells [32,33,34,35,36]; (iii). Reverse immunosuppressive cells [14,37,38,39,40]; (iv). Improve TME microenvironment (for hypoxia, low pH, vasculature, etc.) [41,42,43,44]. The application of nanomaterials in unilateral cancer treatment (e.g., chemotherapy, immunotherapy) has been reviewed [12,45,46,47]. In this paper, we highlight the various types of nanoparticles and their applications as antitumor agents and in regulating the TME. In addition, the clinical application outlook and challenges of these nanoparticles are also discussed.

## 2. The Nanocarriers for Cancer Targeting Therapy

Nanocarriers for cancer therapy can be classified into many types, including inorganic nanoparticles (inorganic NPs), liposome, polymer NPs, biomimetic NPs, and natural NPs (Table 1). They have achieved good treatment effects by stimulating different anticancer mechanisms through delivering of delivery of various therapeutic agents in vivo. Some of them have entered different clinical phases and even been approved as effective drugs by the US Food and Drug Administration (FDA). Most are based on liposome and polymer nanocarriers, and some metal nanocarriers are also approved for imaging diagnosis in clinical practice.

### 2.1. Inorganic NPs

The inorganic nanoparticles include gold NPs, silver NPs, and silica NPs, etc., and among them, gold nanoparticles were the first used for anti-tumor therapy [48,49,90,91]. However, despite concerns for the unavoidable biological safety in vivo, most gold/silver nanoparticles are approved for imaging and diagnosis of cancer in clinical application [92,93]. The biological toxicity of AuNP/Silver NP mainly depends on their physical properties and surface chemical toxicity [94]. In recent years, green surface modification was applied for their biosafety improvement (Figure 1) [48,49,95]. For example, Pandey’s team designed gold nanocages coating with a poly(ethylene glycol) monolayer. The photosensitizer was noncovalently encapsulated in the gold nanocages. These gold nanocages achieved drug accumulation in the tumor site and significantly inhibited the tumor growth with almost no toxicity and phenotypical changes in mice [48]. Park’s team synthesized silver core/shell nanoparticles modified with polyethylene glycol bovine serum albumin (BSA), which has a high indocyanine green (ICG) loading efficiency. The NPs could accumulate at the tumor site. After laser irradiation, the tumor surface temperature rose to 50 ℃ (required for light ablation), the melanoma growth was successfully inhibited, and no obvious systemic toxicity was observed [96]. Previously, a variety of medicinal plant extracts were used to synthesize stable gold or silver NPs with multiple functions as antioxidant, antibacterial, anti-tumor, catalytic, and other biological activities [97,98,99,100,101,102,103]. This might be an effective strategy to promote approval of metal NPs for clinical tumor treatment in the future. 

The application of mesoporous silica nanoparticles (MSNPs) in the biological field successfully demonstrated its advantages of good biocompatibility, high specific surface area and pore volume, allowing large drug loading, and easy chemical modification [104,105,106]. Kumar’s team fabricated folate or N-acetylglucosamine functionalized mesoporous silica NPs encapsulating doxorubicin (DOX-FA-MSNPs or DOX-NAG-MSNPs) for targeted breast cancer therapy [107]. These NPs greatly enhanced the cellular internalization and drug cytotoxicity, which showed little toxicity to normal cells. Engineered MSNPs have widely been employed for cancer drug delivery or imaging. In these systems, many types of molecules and therapeutic agents could be loaded into the nano porous structure or connected onto the surface using different linkers. In any case, controllable diameter, porosity, structure, or chemical composition were combined with selectable properties (e.g., pH, optical, thermal, optical, or magnetic stimulation) for molecular recognition and targeting treatment. 

### 2.2. Lipid Nanocarriers

Liposomes (LPs) are composed of monolayer/multilayer amphiphilic membranes of natural or synthetic lipids [108]. LPs can load hydrophilic drugs in the water core and hydrophobic agents in the lipid bilayer, which makes them flexible and excellent delivery vehicles [109]. LPs are one of the earliest nanoplatforms used in drug delivery systems to carry various active ingredients for cancer treatment. Some liposomal formulations have been approved for clinical trial [57,62,109]. Currently, some LPs loading doxorubicin and paclitaxel which show better treatment for metastatic ovarian cancer and breast cancer in clinical practice have been approved by the FDA [110]. The liposomal nanocarriers could also improve the bioavailability and pharmacodynamics of drugs with poor solubility; however, the lower stability and uncontrollable drug release behavior in vivo affected their wide applications in clinical practice. Some targeting molecular modified liposomal nanocarriers have been designed for precise anti-cancer therapy (Figure 2) [111,112,113,114,115,116,117,118]. For example, Li’s team described a liposome carrier, namely folate (FA) modified liposome (FA-LP) NPs, which could co-deliver erastin and (metallothionein 1D) MT1DP to the cancer location. These LPs could sensitize erastin-induced ferroptosis, decrease cell GSH level, and increase lipid reactive oxygen species (ROS), synergistically inhibiting lung cancer cell growth [117]. The dual targeting strategy might also contribute to promoting LPs’ ability to target tumor cells. Octreotide-modified magnetic liposomes (OMlips) were used for oleanolic acid (OA) loading to form OA-Olips. The LPs exhibited better antitumor effect with little biotoxicity [116]. This kind of targeted molecule and magnetic field-mediated dual-targeted nano-carrier shows great clinical application potential. It was well known that drug resistance at tumor sites are a great obstacle in tumor therapy [112]. It was reported that direct targeting mitochondria is an effective strategy that has been developed in recent years [112,113,118]. Hu’s team developed liposomes with mitochondria target ability for doxorubicin loading from a berberine derivative, which not only increased the drug distribution in tumor, but also achieved better treatment efficiency in tumor-bearing mice with drug resistance [111]. 

In conclusion, although LP nano carriers have been widely studied because of their broad application prospects, they still have limitations including high cost, poor stability in vivo, weak organelle targeting capacity, and easy elimination by phagocytes [112,119,120,121]. Therefore, the development of more accurately targeted and stable liposomes still requires continuous efforts. 

### 2.3. Polymer Nanocarriers

Polymer nanocarriers are widely used in anti-tumor’s agent delivery because of their excellent properties, such as biodegradability, biocompatibility, colloidal stability, low inflammation and immunogenicity, small size, and functionalized surface [122]. However, the polymer nanocarriers still remained uncontrolled in bioavailability and drug release at the tumor site. More and more polymer nanocarriers with the capacity of targeting and stimuli-response were designed for cancer therapy [123,124,125,126,127]. Enhanced permeability and retention (EPR), ligand receptor, polypeptide-mediated tumor targeting and pH, enzyme, hypoxia, and light-response are often considered in the design of nanocarriers (Figure 3). For example, He’s group designed a method to target tumor sites and pH-responsive polymer nano micelles with zwitterionic segments, employed for doxorubicin (DOX) trapping in the hydrophobic layer. The nano micelles showed smaller sizes and high stability in the system circulation and continued to release drugs in the low pH environment via EPR effect-mediated tumor targeting [124,125]. In addition, Hou’s team designed a dual-responsive polymeric nanoparticles, using triethylamine (TEA) as an acid binder; hexachlorocyclic-triphosphonitrile (HCCP) derived cysteine derivatives (CysM) oligomer was polymerized with DOX. Nanoparticles can target the tumor site via the EPR effect and respond to pH and glutathione for releasing an anti-cancer agent. The NPs show the stability of long circulation in blood circulation, but the response to low pH acidic environment in TME makes the drug release rapidly [128]. In addition, a polymeric indoleamine-(2,3)-dioxygenase (IDO) inhibitor based on the poly(ethylene glycol)-b-poly(L-tyrosine-co-1-methyl-D-tryptophan) copolymer (PEG-b-P(Tyr-co-1-MT)) was developed for facile trident cancer immunotherapy [129]. The polymeric IDO inhibitor was modified by Cyclo (Arg–Gly–Asp–D–Tyr–Lys) peptide (cRGD), which can bind to α_v_β_3_ intergrin for targeting tumor cells. Moreover, the polymeric IDO inhibitor can delay the metabolism of l-tryptophan (TRP) to L-kynurenine (KYN) in cancer cells due to the degradation of enzyme responses. In melanoma-bearing mice, DOX in drug-loaded nanoparticles significantly increased matured DCs, CD8^+^ T cells, IFN-γ, and TNF-α, while reducing Treg cells and downregulating PD-L1 expression; this resulted in the improvement of the TME, suppression of tumor, and prolongation of the survival rate. 

Previously, in order to improve the reaction rate and make the drug completely released to the target site, the dual response nanocarriers have been developed to respond to the combination of two signals (e.g., pH/temperature, pH/redox, and photo/temperature) [129,130,131,132]. Therefore, these novel double response nanocarriers have been proved to be anti-cancer drug delivery platforms that can be used for drug control release and targeting of tumor sites, which has a very favorable prospect for the treatment of solid tumors.

### 2.4. Hybrid Nanocarriers

Single-material nano delivery systems are limited to further research and clinical anti-tumor applications due to their inevitable defects, such as low biosafety of inorganic nanoparticles, high cost of liposome nanoparticles, poor targeting, and low bioavailability of polymer nanoparticles [92,93,120,121,125,126]. To solve these problems, a hybrid nanocarrier was developed for delivering anti-tumor therapeutic agents. The nano hybrid system is a nanocarrier that combines organic or inorganic nanomaterials and biological macromolecules into a single composite material (Figure 4). The combination of different nanomaterials can not only make the hybrid nanocarriers show better biosafety and more stable targeting, but also improve the delivery efficiency and bioavailability of drugs [133,134,135,136]. There have many reports about the use of hybrid NPs in cancer detection and treatment. There are some hybrid NPs for clinical diagnosis of cancer, including hybrid magnetic silicon dioxide NPs [137] and hybrid supermagnetic iron oxide NPs (SPIONS) [138,139,140,141]. Some hybrid NPs are used for cancer treatment, such as lipid coated polymers [142,143] and hybrid NPs coupled with genes [144].

One of the reasons why inorganic nanoparticles are mainly used in clinical diagnosis and imaging, but cannot be used in cancer treatment, is due to the lack of surface active groups and difficult surface modification, leading to poor targeting and serious liver toxicity [48,90,91]. However, photothermal effects are a major advantage of inorganic nanoparticles for cancer therapy, while the easy surface modification is the advantage of liposomes or polymer NPs. Therefore, combining the advantages of the two to construct hybrid nanocarriers can not only improve the targeting ability of nanocarriers, but also increase the anti-tumor effect of drugs. For example, Elhabak’ team developed a trastuzumab (TZB) surface modified poly(lactic-co-glycolic) acid (PLGA), circulating NPs that co-encapsulated magnolol (Mag) and gold NPs [139]. The optimized NPs have small particle sizes and high encapsulation efficiency. The surface modified NPs with TZB can target breast cancer due to the over expression of human epidermal growth factor-2 (HER2). The gold NPs and DOX encapsulated in PLGA NPs and DOX showed their photothermal effects and cytotoxicity, respectively, resulting in the multifunctional anti-breast cancer effect. 

Lipid–polymer hybrid NPs are core shell nanoparticle structures comprising polymer cores and lipid/lipid polyethylene glycol (PEG) shells; these hybrids combine physical stability of polymeric nanoparticles and biocompatibility of liposomes. The development of lipid–polymer hybrid NPs broke the limits of single drug delivery and single function design for anticancer therapy. It not only delivers genetic materials, vaccines, and diagnostic imaging agents, but also deliver dual-drugs and targeting design [64]. Fraix et al. reported a lipid–polymer hybrid NP that delivers nitric oxide (NO) and DOX under visible light control [145]. They designed the hybrid nanosystem with DOX entrapped in the PLGA core and a NO photodonor (NOPD) in the phospholipid shell to avoid their mutual interaction. The release of NO inhibits the efflux transporters mostly responsible for DOX cellular extrusion, increasing DOX uptake by cells, and thus enhancing its antitumor activity. 

In conclusion, combining multimodal components in a single hybrid NP allows the structural and functional properties of the resulting NPs to be adjusted in the desired way. The hybrid NPs can enhance their anti-tumor functional properties, and have advantages such as lower cost, simple preparation, good biological safety, precise targeting, controlled drug release, and environmentally responsive drug release, making them more suitable for clinical application.

### 2.5. Biomimetic and Natural Nanocarriers

Nanoparticles (NPs) are becoming more and more common in anti-tumor drug delivery research because they have significant advantages in anti-tumor efficacy and system safety compared with current clinical treatment and diagnosis models [67,69,106]. However, due to its non-specific interaction with phagocytes in vivo during delivery, its clinical application is unsatisfactory [14,25,146]. The retention of NPs by reticuloendothelial system in blood circulation is one of the main obstacles that almost all platforms must overcome. In order to reduce the non-specific interception of NPs, the addition of specific-targeted modification can achieve targeted delivery of NPs, enabling drugs to accumulate at specific sites, which is also an effective strategy to improve drug efficacy of anti-tumor agents. Therefore, biomimetic nanocarriers or natural nanocarriers, an emerging nanotechnology, were previously developed [147,148,149,150]. As the basic unit of biology, cells have a wide range of functions, including the ability to interact with the surrounding environment. They can decrease nonspecific interactions while increase specific targeting (Figure 5) [150,151,152,153,154]. Since 2011, cell membrane coating technology has been developed, for example, in coating the whole red blood cell (RBC) membrane on the surface of NPs [155]. They reported a natural bionic method of particle functionalization, which endows NPs with the function of long cycle delivery by wrapping natural RBC on the surface of biodegradable polymeric NPs. The in vivo results showed that the erythrocyte-mimicking NPs revealed superior circulation than particles without RBC. In 2018, by coating a cancer cell membrane (CCM) on the surface of SPIO@DOX-ICG (superparamagnetic iron oxide @DOX-ICG) nanoparticles, the researchers designed a bio-inspired biomimetic nano system that combines chemotherapy, hyperthermia, and radiation to achieve precise cancer treatment [156]. CCM retains tumor adhesion molecules and surface antigens, making the nano system have tumor homing ability and high biocompatibility. Nano systems can target tumor sites and achieve synergistic anti-cancer effects after systematic administration, without systemic toxic and side effects. Research reported that macrophage, neutrophils, T-cell, platelets, and cancer can all be used as biomimetic materials for coating nanoparticles, which play different targeting delivery and anti-tumor functions [155,156,157,158,159]. Extracellular vehicles (EVs) are also a very safe bionic carrier for drug targeting delivery which can pass through various obstacles without any side effects [158,160,161]. In 2019, Zhu’s team confirmed the anti-glioblastoma (GBM) effect of embryonic stem cells (ESCs) exosome. They then prepared cRGD-modified and paclitaxel (PTX)-loaded ESC-exosomes [162]. The in vitro/in vivo results showed that natural nanocarriers significantly improve the anti-tumor effects of PTX in GBM via an enhanced targeting rate.

In conclusion, biomimetic NPs are a new type of nanocarrier combining the advantages of natural and artificial nanomaterials. Nanoparticles wrapped in cell membranes essentially mimic the characteristics of source cells, giving them a wide range of functions such as long circulation and disease related targeting. Over time, the effectiveness of the cell membrane coating method will undoubtedly expand; there is an inestimable potential for future anti-tumor clinical applications.

## 3. Nanocarriers for TME Regulation and Cancer Therapy

The TME is a highly complex environment that surrounds tumors [163]. More and more evidence shows that TME plays an essential role in controlling tumor occurrence, metastasis, and drug resistance [14,18,72,164]. 

The tumor microenvironment includes the tissue and cellular population surrounding the tumor cells, such as immune cells, endothelial cells, fibroblasts, neurons, and others. These tissues and cellular populations interact with tumor cells, forming a network of cell-to-cell and cell-to-matrix interactions, known as the tumor microenvironment [165]. At the same time, the pH value, the degree of oxygen enrichment, and the redox condition in the tumor site are significantly different from the normal tissue site due to the influence of tumor cell proliferation. The abnormal cell survival conditions further inhibit the activity of immune cells in the tumor site and promote the proliferation and migration of tumor cells [166].

The complexity of TME limits anti-tumor treatment [12,167,168,169]. Therefore, overcoming TME barriers is essential for the deep delivery of therapeutic drugs and treatment effects. Regulating or reprogramming immunosuppressive TME plays important role in cancer therapy. 

Previously, nanomaterials with TME as a target have been widely employed for anticancer drugs delivery to directly regulate the TME, which have achieved promising results [12,164,168]. There are certain ways to reprogram the TME through nanoparticles, such as (i) NPs for destroying extracellular matrix (ECM); (ii) NPs for activating immunostimulatory cells; and (iii) NPs for regulating immunosuppressive factors (Figure 6). In addition, the tumor microenvironment can be improved by adjusting the pH value, oxygen content, and redox environment of the tumor site.

### 3.1. Nanocarriers for Regulating ECM

The extracellular matrix (ECM) of TME is a network of collagen and hyaluronic acid which contains tumor growth factors, anti-inflammatory cytokines, and the tumor vascular system [170,171,172]. In solid tumors, the ECM is considered as a protective chamber that provides a safe environment for the occurrence and development of malignant tumors. Integrin, which transmits information with ECM to inhibit some immune cells and fibroblast’s function, is highly expressed on tumor cells and vascular endothelial cells [173,174]. Furthermore, the dense ECM forms an environment with high-pressure, greatly reducing the deep penetration and diffusion of anti-cancer drugs to weaken the anti-cancer treatment effect [175,176]. Destroying the ECM is a first effective strategy to diminish the barriers of TME (Figure 6) [167,168,172,177,178].

In order to overcome the barrier from ECM, two kinds of nanoparticles were used to destroy the tumor protective effect of ECM: (i) directly destroy the composition of ECM by using collagen nanoparticles; (ii) downregulate the substance expression in the ECM by using enzyme-carrying nanoparticles. For example, Zhou’s team reported NPs with the capacity of degrading hyaluronic acid; they loaded recombinant human hyaluronidase PH20 (rHuPH20) on the surface of PLGA-PEG NPs, and then, modified it with relatively low-density PEG layer to reduce rHuPH20 exposure and prevent it from being removed by macrophages [179]. The facile surface modification reduced TAF activity (an important component of tumor ECM synthesis and remodeling), increased the accumulation of these NPs in 4T1-bearing mice, and inhibited the development of invasive 4T1 tumors at low doses. In addition, matrix enzymes (such as hyaluronidase), matrix metalloproteinase (MMPs), especially matrix metalloproteinase-2 (MMP2), and matrix metalloproteinase-9 (MMP9) have been modified in NPs to destroy the structure of ECM [180,181,182]. Ji’s team assembled the amphiphilic peptide of MMP-2 reaction and phospholipid to construct MMP-2 reactive peptide hybrid-liposome (MRPL) [182]. The pirfenidone (PFD) loaded MRPL can precisely release PFD at the site of the pancreatic tumor and down-regulate the various elements of the ECM by taking advantage of the MMP-2-rich pathological environment. As a result, gemcitabine is more able to penetrate and diffuse into tumor tissues, improving its ability to cure pancreatic cancers. Visibly, modification of NPs with related proteases that destroy ECM components can improve the diffusion of drug-loaded nanocarriers in tumors [173,178]. This is essential for the targeted delivery and release of anti-tumor drugs which may have better anti-tumor effects in combination with other strategies to improve the TME.

### 3.2. Nanocarriers for Activating Immunostimulatory Cells

Due to the complexity of the TME, the cloaking and mutation of tumor antigens lead to immune escape, resulting in immunostimulatory cells losing anti-tumor functions [83,166]. Previously, by using nanoparticles in combination with cancer vaccines, exogenous antigens, immunogenic cell death (ICD) inducers, and immune checkpoint regulators, antigen-presenting cells (antigen presenting cells (APCs), such as dendritic cells (DCs)) and T cell activity can be regulated to improve the local anti-tumor immunity of the TME (Figure 6). Therefore, activating immunostimulatory cells is the second effective strategy to reprogram the TME.

For activating DCs. Nanoparticles can deliver some tumor treatment drugs to tumor cells, causing the ICD effect of tumor cells, realizing tumor cells apoptosis, and transforming tumor cells into anti-tumor vaccines [183]. When ICD occurs in tumor cells, dead tumor cells calreticulin (CRT) will be exposed to the cell surface. At the same time, adenosine triphosphate (ATP) and high mobility group protein B1 (HMGB1) will be secreted and released. They will act on DCs and activate the antigen presentation function, thus activating the anti-tumor T cell response [163,184]. ICD-inducers include mitoxantrone and anthracyclines, oxaliplatin, UVC irradiation, radiotherapy, shikonin, bortezomib, cardiac glycosides (CGs), photodynamic therapy (PDT) with hypericin, and so on [185]. Recently, we prepared a biomimetic PLGA-based nanoparticle (NP) to co-encapsulate plumbagin and dihydrotanshinone (IPLB and DIH) [184]. This NP induced an ICD effect of liver cancer cells, activating DCs and T cells’ immune responses, and generating anti-liver cancer chemo-immunotherapeutic effects by remodeling the TME. 

For activating T cells. Nanoparticles coated with PD-1/PD-L1-targeted ligands have become a new drug delivery system which can improve the drug delivery effect, enhance the immune response, and reduce the side effects of tumor treatment [186,187,188]. The nanoparticle-loading anti- PD-1/PD-L1 can effectively enhance the function of T cells. Consequently, the anti-tumor T cells immune response in TME will be activated. Zhang’s team reported that an engineered cell nano vesicle (NVs) presents PD-1 receptors on its membrane, breaking the PD-1/PD-L1 immunosuppressive axis and enhancing anti-tumor T cell immune responses [187]. In addition, indoleamine 2,3-dioxygenase inhibitors were loaded into PD-1 NVs, synergistically destroying the immune tolerance environment in TME. Importantly, PD-1 NVs significantly increase the infiltration of the CD8^+^ tumor, infiltrating lymphocytes (TILs) and directly driving the tumor killing response of CTL.

In conclusion, immunostimulatory cells such as DCs are one key step to activate an anti-tumor immune response; they can present tumor-associate antigen to CTL and secrete cytokines to activate CD4^+^ T or NKs. Therefore, activating immunostimulatory cells is important for enhancing the immune response of the TME.

### 3.3. Nanocarriers for Decreasing/Regulating Immunosuppressive Factors

The immunosuppressive tumor microenvironment is mainly composed of the complex interaction between immune-tolerance cells (e.g., M2-like TAMs, CAFs, MDSCs, Treg, etc.) and immunosuppressive factors (e.g., TGF-β, VEGF, IL-10,IL-4, HIF-α, etc.) with other cells or non-cells. Immune-tolerance cells and immunosuppressive molecules can promote tumor growth by promoting the formation of ECM and angiogenesis. The failure of immune cells (TAMs and Treg) also seriously affects the development of tumor treatment strategies. Therefore, nano therapies to overcome the immune tolerance of the TME is the third effective strategy to reprogram the TME (Figure 6).

For regulating immunosuppressive macrophages. Macrophages within the tumor, also known as TAMs, are a critical regulator of the immunosuppressive TME for immune escape and tumor development [189]. The majority of TAMs present M2 phenotype and produce immunosuppressive factors to support immunosuppressive cells [190,191]. In contrast to M2 TAMs, M1 generate immunostimulatory factors to activate immunostimulatory cells [192]. Thus, approaches used to polarize TAMs from M2 to M1 have demonstrated great potential for reprogramming the immunosuppressive TME. Chen’s team constructed a fibrin gel which encapsulated anti-CD47-loaded calcium carbonate nanoparticles [193]. This nanogel can scavenge H^+^ in the TME, reversing M2-like TAMs to the M1-like phenotype. The delivery anti-CD47 blocks the “don’t eat me” signal of cancer cells, causing macrophages to engulf more cancer cells. Macrophages can also act as professional APCs, delivering tumor-associated antigens to T cells to initiate the anti-tumor effect of CD4^+^ and CD8^+^ T cells. Shi’s team co-encapsulated photosensitizers indocyanine green (ICG) and titanium dioxide (TiO_2_) with or without ammonium bicarbonate (NH_4_HCO_3_) in mannose-modified PEGylated PLGA nanoparticles for the delivery of photosensitizers to endosome/lysosome or cytoplasm of TAMs [194]. They successfully reprogrammed M2-like TAMs to an anti-tumor M1-like phenotype using this NP, demonstrating superior efficiency and efficacy over lipopolysaccharide stimulation. Reprogrammed TAMs lead to changes in the tumor microenvironment, activation of their anti-tumor function, and release cytokines that recruit more CTLs in the tumor tissues and guide T cells to produce anti-tumor immune memory. 

For regulating immunosuppressive MDSCs. The immunosuppression caused by MDSCs in TME involves many aspects, which can suppress the function of TAMs, T cells, and NK cells in the TME by producing immunosuppressive cytokines like VEGF, IL-10, and TGF-β [195]. MDSCs can also produce peroxynitrite (PNT) to alter the chemokines of TME to prevent the infiltration of T cells. The existence of MDSCs is one of the main reasons for the formation of immunosuppressive TME [46]. Therefore, it is a new strategy of targeting MDSCs to provide anti-tumor therapy. A targeted polymeric micellar nano delivery system (SUNb PM) was constructed. Multi target receptor tyrosine kinase inhibitors were encapsulated in it, cooperating with anti-cancer vaccine therapy to treat advanced melanoma [38]. SUNb PM not only increased the infiltration of CTLs and reduced the number of MDSCs and Treg in TME, but also increased the expression of Th1 type cytokines IL-2 and IFN-γ and downregulated the components related to fibroblasts, collagen, and blood vessels (e.g., CD31, α-SMA, and collagen).

For regulating immunosuppressive Treg cells. The CD4^+^ regulatory T (Treg) cells, as a wide range regulators of gene expression, are critical to the recognition and function of immune regulatory T cell subsets [196]. Tregs, through the activities of cell surface molecules (e.g., Foxp3, CTLA-4, CD25, CD39, CD73, and TIGIT), secretion of cytokines (TGF-β, CCR4), and immune molecules (granzyme, cyclic AMP, and IDO) create an immunosuppressive TME [46,197]. Therefore, blocking the expression of functional molecules on Treg or reducing the number of Treg can reshape immunosuppressed TME. Ou’s team developed tLyp1 peptide conjugated hybrid nanoparticles that target Treg cells of TME [198]. The hybrid nanoparticles presented good stability and effective targeting to Treg cells. By inhibiting the phosphorylation of STAT3 and STAT5, they enhance the downregulation of imatinib on Treg cells. Specifically, when combined with CTLA-4, the Treg cells were more reduced. Prolonged survival rate, inhibited tumor growth, and elevated CD8^+^ T cells infiltration were also observed.

For regulating immunosuppressive CAF and other immunosuppressive factors. Many other strategies have been conducted to reprogram the immunosuppressive TME through inhibiting angiogenesis, CAF function, and soluble immunosuppressive factors (TGF-β, CCL-2, and IL-6, VEGF). Cheng’s team reported a therapeutic peptide assembling nanoparticle that can sequentially respond to dual stimuli in the tumor ECM for tumor-targeted delivery and on-demand release of a short D-peptide antagonist of programmed cell death-ligand 1 (D PPA-1) and an inhibitor of idoleamine 2, 3-dioxygenase (NLG919) [199]. By concurrent blockade of immune checkpoints and tryptophan metabolism, the nano-formulation increased the level of tumor-infiltrated cytotoxic T cells and, in turn, effectively inhibited melanoma growth. Hou’s team developed a nano lotion (NE) formula, targeting the delivery of anti-fibrosis drug fraxinellone (Frax) to CAFs as a method to reverse immunosuppressive TME of melanoma [200]. After intravenous injection of Frax NE, significant reductions of CAFs and interstitial deposition were observed. Immunostimulatory cells (NK cells, CTLs) and factors (IFN-γ, TNF-α) were also increased. Immunosuppress cells (regulatory B cells, MDSCs) and factors (TGF-β, CCL-2, and IL-6) were also decreased in the TME. Cecchini’s team reported successful production of molecularly imprinted polymer nanoparticles (nanoMIPs) against human vascular endothelial growth factor (VEGF) [201]. The composite nanoparticles exhibited specific homing towards human melanoma cell xenografts, overexpressing hVEGF in zebrafish embryos. This nanoMIP can deliver anti-angiogenic drugs that inhibit the development of tumors. 

In conclusion, the TME is a determining factor of the anticancer response and can endow resistance to various anti-tumor therapies. In this context, nanomaterials have been shown to alter populations of CAFs, TAMs, regulatory T cells, and MDSCs. Although considerable progress has been made, in order to translate this strategy into clinical trials, nanomaterial based TME modulation must overcome several limitations, including limited tumor tissue penetration, tumor heterogeneity, and immunotoxicity. Combined treatment with traditional treatment, such as surgery, chemotherapy, photodynamic, etc., may be an effective strategy to solve these problems.

### 3.4. Nanocarriers for Regulating Acidic Environment

The acidic environment in the tumor microenvironment has become a hot topic in cancer biology research. Many studies have shown that tumor tissues generally have a lower pH value compared to normal tissues [166]. This acidic environment may play a crucial role in tumor growth and metastasis.

The metabolic processes of tumor cells can lead to the formation of an acidic environment. For example, tumor cells often have a high rate of glycolysis, producing a large amount of lactic acid and thus reducing the pH value of the surrounding tissue. In addition, the high metabolic rate of tumor cells also leads to increased carbonic anhydrase activity, which accelerates the reaction between carbon dioxide and water to generate a large amount of acidic metabolites [202].

The acidic environment can promote tumor cell proliferation and invasion. On the one hand, a low pH value can promote the inhibition of cell apoptosis, thus, increasing cell survival rate. On the other hand, the acidic environment can also promote tumor cell migration and invasion, as the low pH value increases the protease activity on the cell surface, promoting the interaction between cells and the extracellular matrix.

However, the acidic environment can also have a negative impact on tumor treatment. Some studies have shown that a low pH value can weaken the effectiveness of treatment methods such as radiotherapy and chemotherapy [203]. This is because the low pH value reduces the effective concentration of drugs within the cell, thus decreasing the effectiveness of treatment.

Therefore, exploring the effects of the acidic environment in the tumor microenvironment on tumor growth and treatment has become a hot topic in the field of cancer research. By delving deeper into the metabolic processes of tumor cells and the mechanisms underlying the formation of the acidic environment, it is hoped that more effective tumor treatment strategies can be developed, thus improving the effectiveness and prognosis of cancer treatment.

Nanocarriers can improve the acidic environment of tumor sites in multiple ways. One method is to use nanomaterials with acidic degradation properties. In this way, drugs can be released at the tumor site and gradually degrade over time, reducing local acidity. Siriwallee et al. constructed a type of aminohepthamine cyanine based thermal probe (I2-IR783-Mpip) to achieve photodynamic therapy with specific response to acidic environment at tumor site [204]. Jingyi An et al. used OVA as a template to mineralize calcium carbonate and used the acid degradation reaction of calcium carbonate to improve the pH of tumor sites to enhance the effect of immunotherapy [165]. Som et al. believe that it is impractical and non-selective to use alkaline fluid or proton pump inhibitor to improve the acidic environment of tumors. Therefore, they prepared a series of calcium carbonate nanoparticles with different particle sizes to achieve the function of regulating the acidity of tumor sites [205].

Although nano-carriers have the potential to improve the acidic environment of tumor sites, there are still many technical and safety challenges in their application. In the future, further research and development are needed to give full play to the potential of nano-carriers in tumor treatment.

### 3.5. Nanocarriers for Regulating Hypoxia and Redox Environment 

The oxidative–reductive environment at the tumor site refers to the oxidative–reductive state of cells within the tumor tissue; this is an important difference between tumor cells and normal cells. In the tumor site, the cell metabolism rate is higher than that in normal tissue, resulting in high metabolic activity and cell proliferation. At the same time, the blood supply and vascular generation capacity in tumor tissue are relatively low, leading to low oxygen tension and the formation of a hypoxic state [206]. Therefore, improving the hypoxia condition at the tumor site can effectively inhibit tumor proliferation and improve the treatment effect of chemotherapy, photothermal therapy, and other means. High biocompatibility and oxygen solubility make hemoglobin (Hb) and perfluorocarbon (PFC) effective oxygen transporters [207,208]. Tian et al. used hemoglobin nanoparticles coated by cancer cell membrane to improve the hypoxic environment of tumor sites and significantly enhance the chemotherapy effect of doxorubicin [209]. Song et al. constructed a PEG-modified nanoparticle with tantalum oxide (TaO_x_) as the core and PFC as the shell [210]. By taking advantage of the high oxygen load of PFC and the X-ray absorption characteristics of TaO_x_, the improvement of the hypoxia environment at the tumor site and the radiosensitivity of tumor treatment were achieved.

Hypoxia also results in a difference in the oxidative–reductive environment within the tumor tissue compared to normal tissue. In tumor tissue, there are usually higher levels of oxygen and nitrogen free radicals, which can trigger oxidative stress reactions [211]. These free radicals affect biological macromolecules such as proteins, lipids, and DNA through oxidation–reduction reactions, leading to cell damage and death. In addition, tumor cells typically exhibit higher concentrations of glutathione (GSH), the main reducing ligand of a tumor site. The highest concentration of GSH is found in the cytoplasm of tumors (2–10 mmol·L^−1^), which is substantially higher than its extracellular concentration (2–20 μmol·L^−1^). The GSH concentration in tumor tissue is at least four times higher than that in normal tissue.

The oxidative–reductive environment at the tumor site is of significant importance for the development and treatment of tumors. The proliferation and escape of tumor cells are often influenced by the oxidative–reductive environment. By regulating the oxidative–reductive balance within tumor cells, tumor growth and metastasis can be inhibited. Shuang Bai et al. prepared a star-liked polymer, β- CD-b-P (CPTGSH-co-CPTROS-co-OEGMA) (CPGR), that can respond to both the high ROS environment and the high GSH environment at the tumor site. This polymer can realize the synergistic effect of chemotherapy [211]. Weikai Chen et al. have prepared an alginate gel that can use calcium ions at the tumor site to self-crosslink. The gel includes protoporphyrin (PpIX) modified manese oxide (MnO_2_) nanoparticles and buthionine-sulfoximine (BSO), where MnO_2,_ as a catalyst, can produce oxygen, while BSO inhibits GSH synthesis of cells, so as to improve the tumor microenvironment. Xin Guan et al. prepared a flaky inorganic nanoparticle Nb_2_C, which carried TiO_2_ and *l*-buthionine-sulfoximine to inhibit the synthesis of GSH in tumor cells, affecting the microenvironment of the tumor [212].

Tumor cells need a large amount of oxygen and nutrients when they grow and divide, but their blood supply is usually insufficient, resulting in hypoxia in tumor tissue accompanied by an abnormal redox environment. Hypoxia and abnormal redox environment in tumor site will increase the viability of tumor cells and limit the effect of traditional radiotherapy and chemotherapy. Now, more and more advanced nanotechnology has provided some innovative solutions to the problems of hypoxia and abnormal redox environment at tumor sites. The tumor treatment scheme based on the above nanotechnology has broad clinical prospects.

## 4. Conclusions and Outlook

Over the last few years, cancer targeting therapy based on nanotechnology has gained tremendous attention. Cancer immunotherapy is especially expected to be a game changer for modern cancer therapy. Cancer immunotherapy has made the biggest breakthroughs in recent years, including therapies such as immunotherapy with CAR-T and PD-1/PD-L1 antibodies [9,32,188,213]. However, the patient response rates to such creative treatments remain modest, resulting in several preclinical studies and clinical trials that have directed more attention toward the immunosuppressive TME [9,186,214]. 

Abundant immunosuppressive mechanisms in tumors make it difficult to achieve effective therapeutic effect by single therapy. The controlled release and multi-directional carrier properties of targeted nano-platforms can comprehensively inhibit multiple immune pathways, making cancer immunotherapy more effective. Nanomaterial-based modulation of the TME has been studied for its potential to enhance the efficacy of cancer therapy. Nanomaterials that disrupt the ECM and/or tumor vasculature and increase blood perfusion have been developed to increase the penetration and intracellular delivery of anticancer agents. Nanomaterials that modulate DCs, CAFs, TAMs, Treg cells, or MDSCs have been shown to alter the activities and populations of immune cells in the TME. However, the immune-related adverse events from enhanced immune strategy occur frequently. In order to promote the success of clinical transformation, it is necessary to comprehensively evaluate the safety and immune side effects of nanotechnology. 

In addition, nanocarriers interact with different molecules, cells, tissues, and organs as they are transported through the body. Consequently, biological barriers in the body trap most of these nanoparticles, making it extremely rare for them to reach the tumor site [215]. In order to obtain multifunctional drug delivery nanomaterials with stable groups, targeted ligands, and bio-responsive linkers, complex modifications are required. However, this may increase the obstacles to large-scale, causing repeatable production of nanomaterials and unexpected side effects. Therefore, further investigation must be performed to maintain the balance between the therapeutic benefit, the complexity of formulation preparation/scale-up, and the risk of toxicity before nano-immunotherapy can be satisfactorily applied for cancer patients. 

The rational combination of various cancer targeting treatments based on nanotechnology will result in more efficient cancer inhibition and elimination. A combined therapy strategy can inhibit the occurrence and development of tumor from multiple methods and improve the efficiency of eliminating tumors. Therefore, during the design of a nanomaterial, the combination of multiple treatment methods should be considered. In addition, the powerful advantage of ‘mimicking nature’ still overcomes many of the disadvantages of traditional delivery nanoparticles and provides a more effective strategy for cancer treatment. The natural characteristics of cells, such as the enrichment of targeted proteins, long-term circulation in the body, ability to pass through biological barriers, interactions with other cells, and reduced tissue and cell toxicity, effectively protect the drug delivery of nanoparticles and substantially improve the therapeutic effect. With the rapid development of material science, nanotechnology, pharmacology, bioinformatics, and proteomics, biomimetic nanomaterials are expected to change the current medical technology, overcome many obstacles, and provide new horizons for targeted cancer combination therapy.

According to the research papers on nano-carriers and tumor therapy collected as widely as possible in the past 10 years from 2011 to 2022, although the nano-carriers developed so far have played an excellent role in tumor targeted therapy and tumor microenvironment improvement, the application of nano-carriers still faces limitations of biosafety and clinical technology. To fully exploit the potential of nano-carriers in the treatment of tumors, additional research and development will be required in the future.

## Figures and Tables

**Figure 1 jfb-14-00136-f001:**
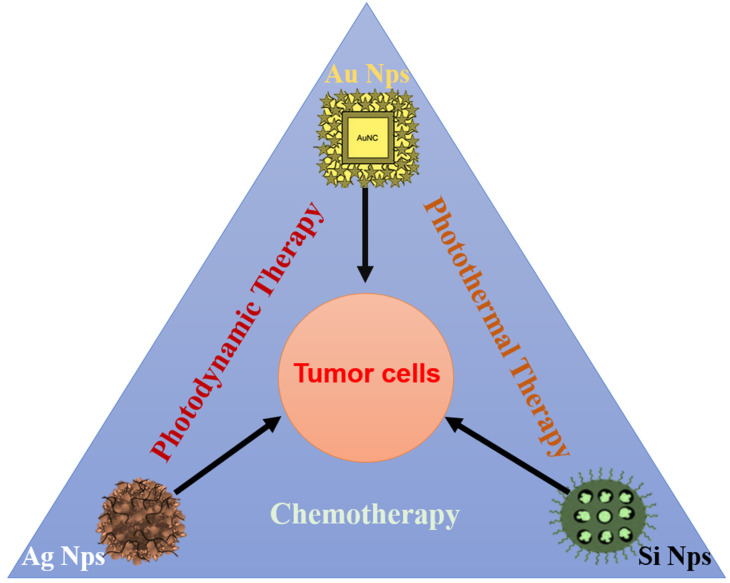
The inorganic nanoparticles (include gold NPs, silver NPs, and silica NPs) were designed for anti-tumor treatment via photodynamic, photothermal, and chemotherapy.

**Figure 2 jfb-14-00136-f002:**
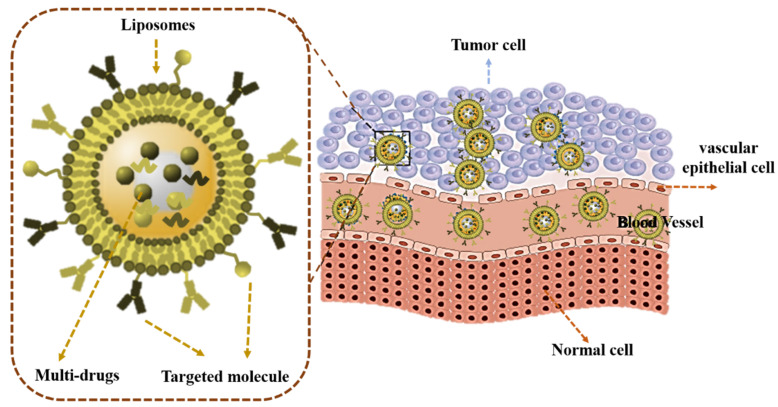
Liposomes co-loading dual-drugs or multi-drugs were prepared for targeting tumor sites and enhancing anticancer effects.

**Figure 3 jfb-14-00136-f003:**
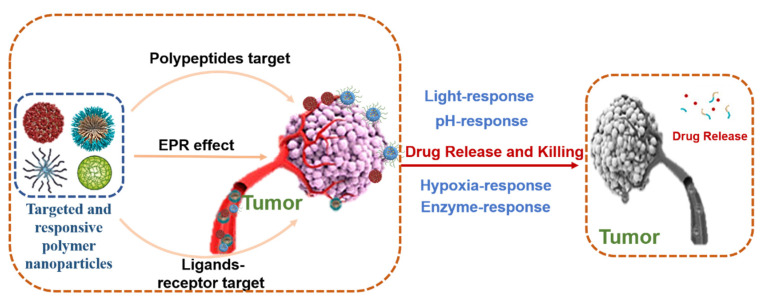
Targeted and responsive polymer nanoparticles design and targeted therapy.

**Figure 4 jfb-14-00136-f004:**
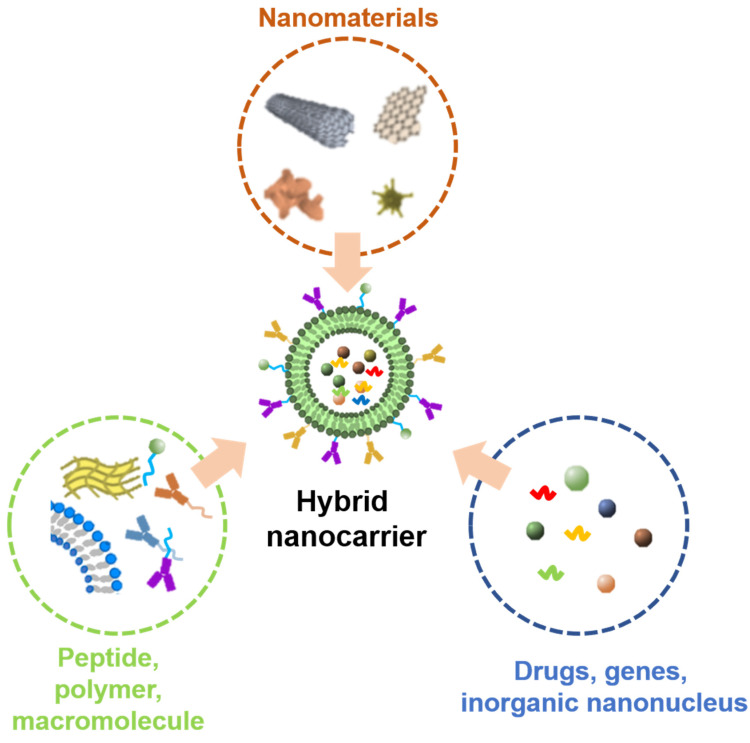
The design and composition of hybrid nanocarriers.

**Figure 5 jfb-14-00136-f005:**
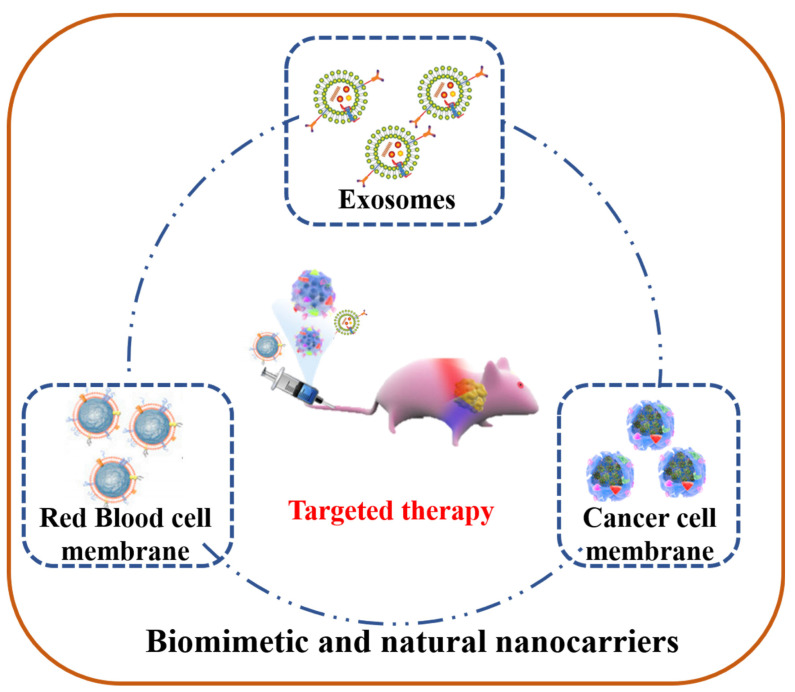
Specific-targeted modified biomimetic and natural nanocarriers for cancer targeting therapy.

**Figure 6 jfb-14-00136-f006:**
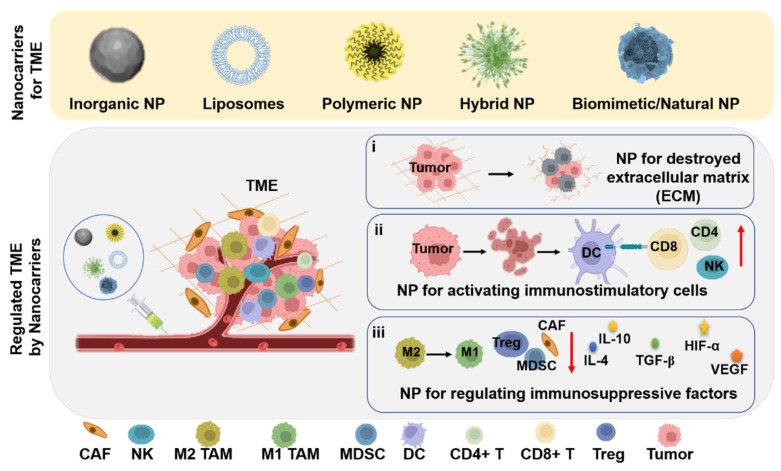
Schematic of nanocarrier-strategies for reprogramming the TME. The strategies of the cancer therapy involved three pathways: (i) NP for destroying extracellular matrix (ECM); (ii) NP for activating immunostimulatory cells; (iii) NP for regulating immunosuppressive factors.

**Table 1 jfb-14-00136-t001:** Development and clinical studies of various types of representative nanocarriers for cancer therapy.

Type	Nanomaterial and Drug	Cancer and Method	Efficacy	Clinical Stage and Ref.
**Inorganic NPs**	Gold and PS	CRC, Breast PTT	Effective delivery and enhances PS’ phototoxicity	Preclinical [48,49]
	Gold and siRNA	GBM Gene therapy	Reduced tumor-associated Bcl2 protein expression	Phase 0 [50]
	Silver and Chinese herb extracts	GBM Undefined	Inhibiting blood vessel formation	Preclinical [51]
	Silver and longan peel powder	Lung Chemotherapy	Down-regulated NF-κB and up-regulated Bcl2, caspase-3 and survival rate of mice.	Preclinical [52]
	Silica and 5-ALA, ZnPc	Skin, Liver PDT	Enhances PS’ phototoxicity, endolysosomal escape	Preclinical [53,54]
**Lipid NPs**	Phospholipid, cholesterol and Doxil, carboplatin, paclitaxel	Kaposi’s Sarcoma, ovarian, breast chemotherapy	Inhibit DNA synthesis and induce apoptosis	Phase 1, phase 3 [55,56,57]
	Phospholipid, Rg3-based liposomes and Ceramide, PTX	HCC, breast immunotherapy	↓ROS and M2-TAM, ↑M1-TAM and CD8^+^T; ↓MDSCs, TME remodeling	Preclinical [58]
**Polymeric NPs**	HPMA, PLGA, PEG-PLA, and DOX, paclitaxel, camptotheci-n	CRC, breast, pancreatic, NSCLCs, ovarian chemotherapy	Inhibit DNA synthesis and induce apoptosis	Phase 1 to 3 [59,60,61,62]
	Folate-PEG-Chems, GalNAc, PEG-b-PCL, PCL-b-PPEEA and Daunorubici-n, siRNA	Leukemia tumor, liver, Pancreatic Chemotherapy, Gene therapy	Enhanced the endocytosis of cells in vitro and in vivo, hepatocyte-specific gene delivery	Preclinical [63,64,65,66]
**Hybrid NPs**	Albumin and Paclitaxel	NSCLCs Chemotherapy	Increase drug solubility, improve bioavailability, and promote absorption of drugs by cancer cells	FDA approved [67]
	PEG, polyglutamic acid, mPEG and D,L-PLA and Camptothecin, paclitaxel	NSCLCs Chemotherapy	Increase drug solubility, improve bioavailability, and promote absorption of drugs by cancer cells	Phase 0 to 2 [68,69,70]
	SPIONs, PNIPAAm-MAA and DOX	Lung cancer Chemotherapy, imaging	pH-dependent manner, time-dependent manner	Preclinical [71]
**Biomimetic NPs**	DC, tumor antigen, and sunitinib	GBM, melanoma, prostate, immunotherapy + radiochemotherapy, immunotherapy + chemotherapy	Activate T-cells’ immune response	FDA approve [72,73]
	Neutrophil, RBC, Platelets and Celastrol, tumor antigen, CpG, R848	Pancreatic cancer, melanoma chemotherapy, immunotherapy	Targeting tumor site, prevent liver metastasis of tumor, improve the survival rate of tumor bearing mice, activate immune response	Preclinical [14,74,75,76,77,78]
**Natural NPs**	Platelets and PD-L1	Melanomas, breast cancer, immunotherapy	Delivery of anti-PDL1 to the surgical bed and target CTCs, reduce the risk of cancer regrowth and metastatic spread	Preclinical [79,80,81,82]
	Macrophages Chemical drugs	breast, GBM chemotherapy	Targeted to cancer cells, inhibit tumor invasion	Preclinical [83,84]
	Exosomes mi-RNA, chemical drugs, proteins	GBM, breast, ovarian cancer Gene therapy, chemotherapy, immunotherapy	Activate T cell response, inhibit tumor growth	Preclinical [85,86,87,88,89]

NPs: nanoparticles; PTT: photothermal therapy; PS: photosensitizer drug; PDT: photodynamic therapy; ROS: reactive oxygen species; HCC: Hepatocellular cancer; PTX: paclitaxel; CRC: colorectal cancer; NSCLCs: non-small cell lung cancers; SPIONs: Supermagnetic iron oxide nanoparticles; PNIPAAm-MAA: radical polymerization of methacrylic acid (MAA) and N-isopropylacrylamide (NIPAAm); GBM: glioblastoma.

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
