# Peer review of "Tumor Microenvironment Regulation and Cancer Targeting Therapy Based on Nanoparticles"

_jfb, 2023, doi:10.3390/jfb14030136_

Round 1

Reviewer 1 Report

Hereby is presented a short review realized on the base of the preliminary version of “Tumor Microenvironment Regulation and Cancer Targeting Therapy Based on Nanoparticles by Han et al, a comprehensive literature review addressing the reader with more or less wide-known concepts and/or data mainly revolving around the notion of Tumor Microenvironment (TME) and nanoparticle-based cancer treatment. TME represents the diversified array of molecular, cellular and physical condition that characterize the intra- and peri-tumoral spaces, besides incorporating various mechanisms that enhance malignant cells to survive and develop resistance to conventional therapeutic strategies. The aforementioned rationale for the study is pragmatically presented in the first, introduction paragraph of the review, together with the evolution of treatment algorhythms for cancer as a whole from immune-based to nanoparticles (NP)-based targeted therapy.

Paragraph 2 and 3 are dedicated to detailing of current availability and diversity of nano-materials and nano-engineering technique, each of them with specific advantages and application that may favor indications or, on the opposite, contraindications based on the subset of malignancy to be addressed for therapy. Each main branch is presented first from a structural and then from either an experimental, preclinical or clinical angle with the current body of evidence coming from the analyzed literature presented separately. Such data are also schematized into a large table that allows the reader to readily fetch and analyze the degree of advancement for each specific nanocarrier separately, from inorganic NP to lipid, polymer, hybrid, biomimetic and natural NP. Moreover, after detailing means of NP-based cancer therapy, mechanism of action are further presented, with a concise repartition into 3 main pathways of action: ECM destruction, regulation of TME-related immunosuppressive factors(immunosuppressive macrophages, MDSC, Treg cells, CAF), activation of immunostimulatory cells(CD and T-cells).

Despite the sufficient degree of clarity in the way such complex data has been illustrated after reviewing the global literature, methods for search, selection and inclusion of the literature itself are missing, therefore inevitably raising bias concern in the process. Still, acknowledgements for funding are transparent as much as disclosure for lack of any conflict of interest. Going past that, conclusions presented in the last paragraph obviously insist on the lack of sufficient data for a potentially revolutionary step into oncological clinical practice, which is right now missing. Further research is warranted in order to fully elucidated the level of biocompatibility of nanoparticles and NP-based oncologic regimens and long term effects on the human body, in order to establish a balance in between disease-specific burden and treatment-specific morbidity on the long run.

Author Response

Answer:

We thank the reviewer very much for taking her/his valuable time to review our manuscript. These positive feedback and encouragement are the biggest inspiration to the authors.

Q1. Despite the sufficient degree of clarity in the way such complex data has been illustrated after reviewing the global literature, methods for search, selection and inclusion of the literature itself are missing, therefore inevitably raising bias concern in the process.

A: we supplemented the main time range of the literature collection to support the author's view in the review. For detailed changes, please see the end of the conclusion section.

Q2. Going past that, conclusions presented in the last paragraph obviously insist on the lack of sufficient data for a potentially revolutionary step into oncological clinical practice, which is right now missing.

A: Based on your suggestions, we rewrote the conclusion section to make sure it has clear opinions.

Reviewer 2 Report

   The review  entitled “Tumor Microenvironment Regulation and Cancer Targeting Therapy Based on Nanoparticles"is generally well written and structured. However, some changes should be done by authors as following  

1-In abstract, authors should replace the word roaring  in line 21with tremendous for example.

2- In line 27, I think TAM should be replaced with TME.

3- in line 80, in antitumor and should be replaced with as antitumor agents and in

4- in line 88 deliver should be replaced with delivering of delivery of and in line 110 accumulate should be replaced with accumulation.

5- using of recently to express recent studies should be for the studies only in the last two years, otherwise replace it with previously.

6- in line 126 for targeted breast cancer therapy would be better than targeting anti-breast cancer

7- Full name of all abbreviations should be added in its first place mentioning within text so please check the whole article for that.

8- in line 148 add practice after clinical.

9- in line 157 which receptor do you mean? Please demonstrate that.

10- After tumor therapy in line 159, the reference should be added.

11-In line 164 replace resistant with resistance.

12- All figure legends must be fully demonstrated for example in figure 2, authors should indicate the different parts of liposomes. Also, all abbreviations full names should be added.

13- In line 176 add and before functionalized surface.

14- in line 177 replace uncontrol with uncontrolled and in line 250 released with release.

15- in line 271 follow antitumor with agents.

16- The part starting from line 358 till line 364 is missing references, please authors add them.

17- in line 375, number of reference should be in the same style of other references in the paper.

18- Another part for targeting hypoxia and elevated GSH in tumor microenvironment using nanoparticles should be added in this manuscript.

Author Response

Answer:

Thanks for your appreciation of our work with instructive suggestions. Based on your suggestions, we rewrote the manuscript to make it more readable. We revised the

manuscript one by one as follows.

Q1. In abstract, authors should replace the word roaring in line 21with tremendous for example.

A: Thank you for reading our manuscript carefully. We have made corresponding changes according to your comments.

Q2. In line 27, I think TAM should be replaced with TME.

A: Thank you for reading our manuscript carefully. We have made corresponding changes according to your comments.

Q3. in line 80, in antitumor and should be replaced with as antitumor agents and in

A: Thank you for reading our manuscript carefully. We have made corresponding changes according to your comments.

Q4. in line 88 deliver should be replaced with delivering of delivery of and in line 110 accumulate should be replaced with accumulation.

A: Thank you for reading our manuscript carefully. We have made corresponding changes according to your comments.

Q5. using of recently to express recent studies should be for the studies only in the last two years, otherwise replace it with previously.

A: Thank you for reading our manuscript carefully. We have made corresponding changes according to your comments.

Q6. in line 126 for targeted breast cancer therapy would be better than targeting anti-breast cancer

A: Thank you for reading our manuscript carefully. We have made corresponding changes according to your comments.

Q7. Full name of all abbreviations should be added in its first place mentioning within text so please check the whole article for that.

A: We are very sorry for the reading trouble caused to you. We have checked all the abbreviations in the manuscript, and added full spelling and explanation in the places where the abbreviations first appeared in the text. And we also have supplemented the corresponding list of abbreviations.

Q8. in line 148 add practice after clinical.

A: Thank you for reading our manuscript carefully. We have made corresponding changes according to your comments.

Q9. in line 157 which receptor do you mean? Please demonstrate that.

A: We have already restated the unclear statements.

Q10. After tumor therapy in line 159, the reference should be added.

A: Thank you for reading our manuscript carefully. We have made corresponding changes according to your comments.

Q11. In line 164 replace resistant with resistance.

A: Thank you for reading our manuscript carefully. We have made corresponding changes according to your comments.

Q12. All figure legends must be fully demonstrated for example in figure 2, authors should indicate the different parts of liposomes. Also, all abbreviations full names should be added.

A: We apologize for the poor quality of the pictures used in the article and the ambiguity of the meaning. We have redrawn Figure 1 - Figure 5 and hope that the problem can be improved.

Q13. In line 176 add and before functionalized surface.

A: Thank you for reading our manuscript carefully. We have made corresponding changes according to your comments.

Q14. in line 177 replace uncontrol with uncontrolled and in line 250 released with release.

A: Thank you for reading our manuscript carefully. We have made corresponding changes according to your comments.

Q15. in line 271 follow antitumor with agents.

A: Thank you for reading our manuscript carefully. We have made corresponding changes according to your comments.

Q16. The part starting from line 358 till line 364 is missing references, please authors add them.

A: Thank you for reading our manuscript carefully. We have made corresponding changes according to your comments.

Q17. in line 375, number of reference should be in the same style of other references in the paper.

A: Thank you for reading our manuscript carefully. We have made corresponding changes according to your comments.

Q18. Another part for targeting hypoxia and elevated GSH in tumor microenvironment using nanoparticles should be added in this manuscript.

A: Your suggestion is very meaningful. We have added some reviews on nano-carriers that regulate tumor acidity, hypoxia and redox environment, and placed them under the subtitles of 3.4 and 3.5 respectively.

Reviewer 3 Report

1. Give proper abbreviations with full spelling at one time when they come in for the first time in the text

2. For a better understanding of the readers the role of TME in tumor initiation, progression, and metastasis should be included. 

Author Response

Q1. Give proper abbreviations with full spelling at one time when they come in for the first time in the text.

A: We are very sorry for the reading trouble caused to you. We have checked all the abbreviations in the manuscript, and added full spelling and explanation in the places where the abbreviations first appeared in the text. And we also have supplemented the corresponding list of abbreviations.

Q2. For a better understanding of the readers the role of TME in tumor initiation, progression, and metastasis should be included. 

A: Thank you very much for your questions about the interpretation of TME in our manuscript. We have added some detailed introduction about TME under the headings of “3. Nanocarriers for TME regulation and cancer therapy” to make readers more clear about TME.

Reviewer 4 Report

The theme of the submitted article will be helpful in context of TME modulation using various formulations. The discussion part with respect to formulations and TME regulation/modulation is not up to date. Therefore, I would recommend major revision.

My comments are as follows:

1. Graphical abstract is missing. I would strongly suggest including a GA.

2. Fig. 1 looks very general and non-informative. Please make it attractive by using suitable graphics and label each entity.

3. I would suggest discussing about large molecules such as peptides and mRNA-based vaccines too.

4. Fig 2. is not in presentable form. Please mark/label all entities to have a better understanding.

5. The discussion part about the nanoparticles should be elaborated and mention the outcome of the formulations in detail. There are couple of more formulations available for TME regulations. Please try to include as much as possible to enhance the overall quality of the article.  

6.  Fig 3. looks like a very basic representation. Please modify and mark/label all entities to have a better understanding.

7.  Similarly, Fig. 4 and 5.

8.  I found that the authors have copied few sentences directly from the published articles, such as: Line 346 – 350 has been copied from the abstract of reference no. 180.

Please go through the manuscript carefully and re-write the sentences to avoid plagiarism.

9.  Provide the list of abbreviations at the end of the manuscript as per the guidelines.

10.  Please discuss about the following parameters in detail and provide sub-heading to each factor responsible for modulating TME such as:

Drug delivery systems to target tumor hypoxia/ targeting tumor vasculature/ formulations to target TAMs/ targeting tumor stromal cells/ tumor micro-acid environment, etc.

Author Response

The theme of the submitted article will be helpful in context of TME modulation using various formulations. The discussion part with respect to formulations and TME regulation/modulation is not up to date. Therefore, I would recommend major revision.

A: Thanks for your kind suggestion. We apologize for the unsatisfied parts of the manuscript. We will revise it carefully according to your comments.

Q1. Graphical abstract is missing. I would strongly suggest including a GA.

A: We added a graphical abstract to the manuscript.

Q2. Fig. 1 looks very general and non-informative. Please make it attractive by using suitable graphics and label each entity.

A: We apologize for the poor quality of the pictures used in the article and the ambiguity of the meaning. We have redrawn Figure 1 - Figure 5 and hope that the problem can be improved.

Q3. I would suggest discussing about large molecules such as peptides and mRNA-based vaccines too.

A: I think there should be further discussion on your suggestion. The main content of our manuscript is to describe the application of nano-carriers, including their functions of targeted tumor treatment and tumor microenvironment improvement, and to classify the types of nano-carriers. Therefore, the main purpose of this article is to discuss the nano-carriers, not the material carried by the nano-carrier. Peptides, RNA and other substances are only media used to achieve multiple functions of nano platform. However, in the manuscript, we have also described the relevant contents of large molecules, which are distributed in various parts of the article. For example, the description of polypeptide: “In 2019, Zhu’s team confirmed the anti-glioblastoma (GBM) effect of embryonic stem cells (ESCs) exosome and then prepared cRGD-modified and paclitaxel (PTX)-loaded ESC-exosome[162].”, “Ji’s team assembled the amphiphilic peptide of MMP-2 reaction[180]”, “Ou’s team developed tLyp1 peptide conjugated hybrid nanoparticles that target Treg cells of TME[196].”, “Cheng’s team reported a therapeutic peptide assembling nanoparticle that can sequentially respond to dual stimuli in the tumor ECM for tumor-targeted delivery and on-demand release of a short D-peptide antagonist of programmed cell death-ligand 1 (D PPA-1) and an inhibitor of idoleamine 2, 3-dioxygenase (NLG919)[197]”. And more relevant contents not listed and their references: [3], [36], [197].

The description of mRNA written in the article is not expanded here. The serial numbers of relevant references are as follows: [28], [29], [32], [50], [64], [66], [86], [88], [112], [143], [144], [157], [160].

Q4. Fig 2. is not in presentable form. Please mark/label all entities to have a better understanding.

A: Same as question 2, we have redrawn Figure 1 - Figure 5 and hope that the problem can be improved.

Q5. The discussion part about the nanoparticles should be elaborated and mention the outcome of the formulations in detail. There are couple of more formulations available for TME regulations. Please try to include as much as possible to enhance the overall quality of the article. 

A: Thanks for your kind suggestion. We have supplemented the corresponding contents.

Q6. Fig 3. looks like a very basic representation. Please modify and mark/label all entities to have a better understanding.

A: Same as question 2, we have redrawn Figure 1 - Figure 5 and hope that the problem can be improved.

Q7. Similarly, Fig. 4 and 5.

A: Same as question 2, we have redrawn Figure 1 - Figure 5 and hope that the problem can be improved.

Q8. I found that the authors have copied few sentences directly from the published articles, such as: Line 346 – 350 has been copied from the abstract of reference no. 180.

Please go through the manuscript carefully and re-write the sentences to avoid plagiarism.

A: We carefully examined the full text of the manuscript and restated the description of the problem.

Q9. Provide the list of abbreviations at the end of the manuscript as per the guidelines.

A: Thanks for your kind suggestion. We have supplemented the corresponding tables.

Q10. Please discuss about the following parameters in detail and provide sub-heading to each factor responsible for modulating TME such as:

Drug delivery systems to target tumor hypoxia/ targeting tumor vasculature/ formulations to target / targeting tumor stromal cells/ tumor micro-acid environment, etc.

A: We have added some reviews on nano-carriers that regulate tumor acidity, hypoxia and redox environment, and placed them under the subtitles of 3.4 and 3.5 respectively.

Round 2

Reviewer 4 Report

The authors have reasonably addressed most of my comments. Therefore, I would recommend its publication after minor revision.

Please check abbreviation list - Rows and columns are not properly aligned. 

Author Response

Responses to Reviewers’ Comments

Reviewer #4

Comment:

The authors have reasonably addressed most of my comments. Therefore, I would recommend its publication after minor revision.

Q: Please check abbreviation list - Rows and columns are not properly aligned. 

A: Thank you again for your careful reading of the manuscript. We carefully checked the abbreviation table and matched each abbreviation with the complete spelling.
